# Investigation of an outbreak of acute algal-associated dermatoses among artisanal fishermen in Senegal: A one health approach

Mbouna Ndiaye[1]*, Ndeye Magatte Ndiaye[2], Fatimata Ly[3], Mamadou Ndiaye[4], Boly Diop[4], Diambogne Ndour[5], Alassane Ndiaye[6], Elhadji Mamadou Ndiaye[4], Moussa Diallo[7], Mamadou Fall[8], Marie Khemesse Ngom Ndiaye[9]

**1** Field Epidemiology Training Program, Ministry of Health and Social Action, Dakar, Senegal, **2** Regional Health Directorate of Dakar, Dakar, Senegal, **3** Department of Dermatology-Venerology, Institute of Social Hygiene, Cheikh Anta Diop University of Dakar, Dakar, Senegal, **4** Directorate of Prevention, Ministry of Health and Social Action, Dakar, Senegal, **5** Health Emergency Operations Center, Ministry of Health and Social Action, Dakar, Senegal, **6** Health and Sustainable Development Training and Research Unit, Alioune Diop University of Bambey, Dakar, Senegal, **7** Dermato-Pathology Laboratory, Cheikh Anta Diop University of Dakar, Dakar, Senegal, **8** Poison Control Center, Cheikh Anta Diop University of Dakar, Dakar, Senegal, **9** Directorate General of Health, Ministry of Health and Social Action, Dakar, Senegal

* mbounandiaye@hotmail.fr

## Abstract

In November 2020, an alert for a "mysterious disease" among fishermen was issued. Fishermen are particularly subjected to dermatoses due to their constant contact with seawater, fish, crustaceans, and fishing equipment that may contain harmful agents. The study aimed to examine the alert, identify the causative agent and suggest preventive and control measures. This was a cross-sectional study of dermatoses in Dakar (Senegal) from October 11 to November 30, 2020, using quantitative and qualitative methods within a 'One Health' approach." The investigation included bacterio-virological, anatomopathological and toxicological examinations. Data were analyzed using Epi info and QGIS (case mapping), We observed all confidentiality measures during the study. A total of 555 cases were diagnosed with an attack rate of 5.4% among fishermen and no deaths were reported. There was a delay in epidemic detection and notification. The epidemic was most prevalent among people from coastal areas. Average age of cases was 22 ± 9 years, and all were male and artisanal fishermen by profession. Patients presented with fever (16%), cutaneous pain (100%) and mucocutaneous lesions (100%) consisting of vesicles, papules and ulcerations localized on exposed areas of the body, external genitalia and oral mucosa, with severe cases (8%). Toxicology revealed the presence of a toxic alga (*V. rugosum*) in marine equipments. The notion of a sea trip in the 24–48 hours before the onset of the disease was found in 92%. Majority of cases (74%) did not have full personal protective equipment (PPE). The proportion of people without full protection was 83% among those who developed severe forms. People without full protection

**Data availability statement:** We declare that all the raw data and processing files used for the analysis presented in this article are available within the paper and its Supporting information files. Any additional request for information can be met on the condition that it is never disseminated without being anonymized or without the opinion of the authors. While remaining available for any clarifications, we ask you to accept the expression of our distinguished sentiments.

**Funding:** The authors received no specific funding for this work.

**Competing interests:** The authors have declared that no competing interests exist.

were more exposed to severe forms than those with full PPE; (OR = 1.818; 95% CI [0.829 - 3.988]). The investigation has linked the epidemic to a probable algal origin. We need to promote the use of personal protective equipment and improve the early warning and notification system.

## Introduction

The interaction between the environment and human health is a major concern because pollution, whether airborne, waterborne, or terrestrial, is increasingly frequent on a global scale, causing acute events or chronic diseases.

The World Health Organization estimates that environmental problems are the cause of 21% of diseases worldwide [1]. The vast majority of these diseases occur in developing countries, and the proportion attributable to environmental causes is higher in low-income countries. Among these pathologies, acute dermatoses occupy a prominent place as they are common conditions in medical practice in tropical areas with multiple aetiologies: infectious agents (bacteria, viruses, parasites), mechanical, physical, chemical, etc. Hundreds of new agents are introduced into workplaces each year, many of which can damage the skin either as primary skin irritants or as sensitizing and allergenic substances [2]. For example, fishermen are particularly subjected to dermatoses due to their constant contact with seawater, fish, crustaceans, and fishing equipment that may contain harmful agents.

Statistics on dermatoses in the workplace vary greatly from one study to another and from one country to another. The literature in this field is scarce and studies are often fragmented. In Europe, 20–34% of diseases occurring in the workplace are occupational dermatoses [3]. In France, they represent about 10% of general skin pathology and affect 1% of all professionals [4]. In the United States, the prevalence of dermatitis among workers is 9.8% overall, with rates ranging from 5.5 to 15.4% depending on the sectors of activity [5]. In Tunisia, dermatological pathology accounts for 4% of declared occupational diseases [6]. In a study in Dakar (Senegal) on contact dermatitis, the predominant professional sectors were construction, mechanics, health, cleaning, and commerce [7].

Dermatoses caused by marine algae such as *Vulcanodinium rugosum* are exceptional both in the workplace and outside. To our knowledge, the only study to date in the world was reported in Cuba [8].

In November 2020, an alert regarding a "mysterious disease" among fishermen characterized by multiple dermatoses with particularly severe cases was transmitted to the Mbao health district. Amid public clamor and the dissemination of information relayed by the national and international press [9], an investigative team was urgently dispatched with the primary objective of investigating the dermatoses occurring in the Dakar region. Specifically, the aim was to measure the magnitude and severity of the health concern, determine the causes of the acute dermatoses observed, and establish control and prevention measures.

## Methods

### Study setting

Located at the extreme west of the African continent, Dakar region covers an area of 550 km², which is 0.28% of the national territory. It is bordered to the east by Thies region and to the north, west, and south by the Atlantic Ocean. The population of Dakar is estimated at 3,732,282 inhabitants over an area of 550 km², corresponding to a density of 6,786 inhabitants per km² [10]. Nearly a quarter of Senegal's population (23.2%) lives in Dakar, making it the most populous region in the country. More than three-quarters of the population live in the suburban departments (Pikine, Guediawaye, and Rufisque).

Fishing, commerce, tourism, and crafts represent the main economic sectors locally. Given a combination of circumstances, including successive years of drought and the deterioration of trade terms following the oil shock, fishing quickly became one of the most important economic sectors. With a coastline of 133.69 km in length, the region has 10,200 fishermen, representing 15% of the fishing population in Senegal (68,000 people [11]. Beyond its economic feature, fishing is a culture, a tradition. It is practiced by a community of fishermen using more than twenty fishing techniques, with strategies that vary seasonally based on biological and socioeconomic factors [12]. The artisanal fishing sector makes a remarkable contribution for achieving food self-sufficiency. The artisanal processing of fish products is a labor-intensive sector. It allows the valorization of fish products and facilitates the preservation of fish through drying, smoking, cooking, or fermentation. It absorbs nearly a third of the landings of fresh fish products [13].

However, the conditions under which fishing is practiced involve risks related to marine pollution. Additionally, fishermen do not always have adequate personal protective equipment (boots, tarps, hoods) against hazards due to their relatively high cost. They handle substances and marine products daily, the misuse of which can cause dermatological lesions. Thus, the conditions under which this fishing is practiced pose challenges related to the safety and health of the workers.

In Dakar region, the epidemiological surveillance system is managed by the Regional Health Directorate, which receives signals from the district, constituting the operational level of the health system around which care delivery points, reception, and patient care structures revolve. To ensure early warning and rapid response, the data collection and analysis system of the Integrated Disease and Response Strategy (IDRS) relies, among other tools, on two main information or signal production channels: indicator-based surveillance and event-based surveillance.

### Type and period of study

We conducted a observational and cross-sectional study ranging from October 11, 2020 to November 30, 2020.

### Study population

*Statistical units*: They consist of individuals with acute dermatological lesions observed during the investigation period.

*Case definition*: Any person presenting with papulo-vesiculo-necrotic dermatological lesions localized on exposed areas of the body and/or external genital organs, who had been at sea within the five days preceding the onset of symptoms or had been in contact with canoes or marine equipment.

### Data collection

*Collection technique*: We conducted a documentary review of several information sources from health structures (consultation registries, hospitalization registries), interviews with individuals concerned by the study, their families, healthcare providers, and an observation of the equipment and canoes used at sea. In the diagnostic process, several hypotheses were put forward. Therefore, in addition to the epidemiological study, the in-depth investigation included four components that took into account possible hypotheses, whether microbial, toxicological, or other.

- Bacteriological and virological tests at the Pasteur Institute in Dakar on blood and urine samples and nasopharyngeal swabs from 10 randomly selected patients;

- Anatomopathological examinations (skin biopsy) at the Dermatopathology Laboratory of Cheikh Anta Diop University in Dakar of skin samples from 4 randomly selected patients;

- Environmental study involving 8 random samples taken from fishermen's equipment (nets, gloves, etc.) and other marine products for toxicological testing at the Poison Control Center (Senegal), the Regional Center for Research in Ecotoxicology and Environmental Safety (Senegal), and the French Research Institute for Exploitation of the Sea (IFRE-MER) in Nantes (France). Light microscopy was used to isolate the cells (IX51 inverted microscope (Olympus, Tokyo, Japan) while quantitative PCR analysis was performed to detect the causative agent. Further analysis using liquid chromatography-tandem mass spectrometry (LC-MS/MS) identified specific toxins.

- A qualitative study was conducted to assess the psychological and socioeconomic consequences of the disease on individuals and communities, to identify the strategies and measures taken to control the epidemic, and finally to draw lessons learned from the outbreak. We selected 20 volunteer participants via convenience sampling from patients seen in health services. Individual semi-structured interviews were conducted using an interview guide. Data processing followed a thematic approach through iterative manual coding. Units of meaning were classified into major themes, including socioeconomic impact, COVID-19-related stigma, and therapeutic recourse. This methodology allowed for the triangulation of quantitative data with the lived realities of artisanal fishing actors, thereby strengthening the overall understanding of the psychosocial impact of this health crisis.

*Data collection tools*: We created a standardized form (investigation form) in paper format to collect the variables of interest among the study population.

*Variables of study*: these were the following variables

- *Socio-demographic characteristics*: sex, age, home address, profession.

- *Clinical and paraclinical characteristics*: dermatological lesions, fever, headaches, pruritus, pain, bleeding, ocular lesions, date of disease onset, disease evolution, and blood, urine, and skin tests.

- *Information on exposure:* Sea voyage, date of voyage, travel zone, type of fishing, personal protective equipment, use of decoction, handling of marine products.

*Collection Procedure*: The investigation team met with administrative, local, and health authorities, leaders of fishing organizations, community relays, *"badienoux gox"* (neighborhood godmothers), and all key informants.

The investigators collected data and took samples during the admission of patients to health facilities. Patients admitted to reception services were interviewed using the investigation form created for this purpose. The investigators reviewed registers to search for cases and also took samples from fishermen's equipment and other marine products.

## Data management

We entered and analyzed the data using Epi Info software version 7.2.2. We cleaned and corrected missing data, inconsistencies, and duplicates using the completed questionnaires. QGIS software version 3.18.2 was used for case mapping. We calculated proportions, attack rates, and lethality rates (for qualitative variables) and measures of central tendency and dispersion (for quantitative variables). We described the epidemic in terms of temporal evolution (epidemic curve), location (case mapping), and individuals (individual characteristics). We calculated odds ratios to measure associations and compared proportions using the chi-square test or Fisher's exact test (depending on applicability conditions), with the significance threshold set at $p < 0.05$.

## Ethical considerations

The investigation was commissioned by the Ministry of Health authorities (reference n°000698/MSAS/DGS/DP of 9 October 2020). We obtained free and informed consent from all those who responded to the questionnaires. For subjects aged 18 years or younger, parental or guardian consent was required. We adopted all necessary confidentiality measures for the investigation. The patients' images were masked, anonymized and used with their consent. We provided medical emergency care on-site at health posts and centers.

## Results

Fig 1 shows the selection scheme of the respondents with 4,137 suspected patients consulted in the health facilities at the beginning of the survey.

### Temporal evolution of the outbreak

A total of 555 cases were diagnosed with an attack rate of 5.4% among fishermen and no deaths were reported. Considering the general population of the region instead of the fishermen exposed, this attack rate was 14.8 per 100,000 inhabitants. The onset of the outbreak goes back to October 18, 2020 with a peak occurring November 12, 2020, with 93 cases (Fig 2). An apparent 'decline' in cases was observed between November 13 and 14, 2020, corresponding to the weekend.

   The investigation reveals that the first case received in consultation was on November 12, 2020, at the Thiaroye sur Mer health post (Mbao district). Consequently, the detection date (November 12) occurred 25 days after the onset of the outbreak (October 18).

   Notification to the district was made on November 16 by the Thiaroye sur Mer health post, four days after detection (November 12).

   A series of control measures (actions) were taken starting from November 17, within 24 hours after the notification.

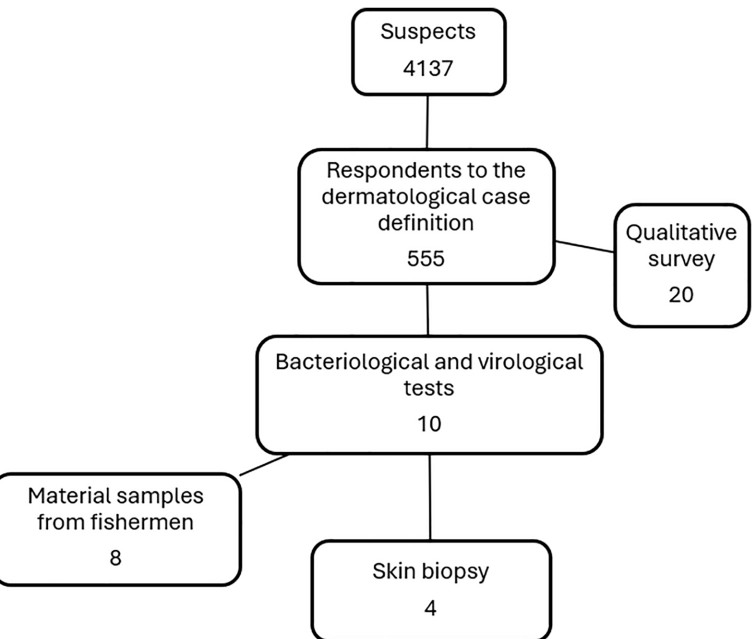

**Fig 1. Patients' selection scheme for the survey of dermatoses observed in fishermen, Senegal**.

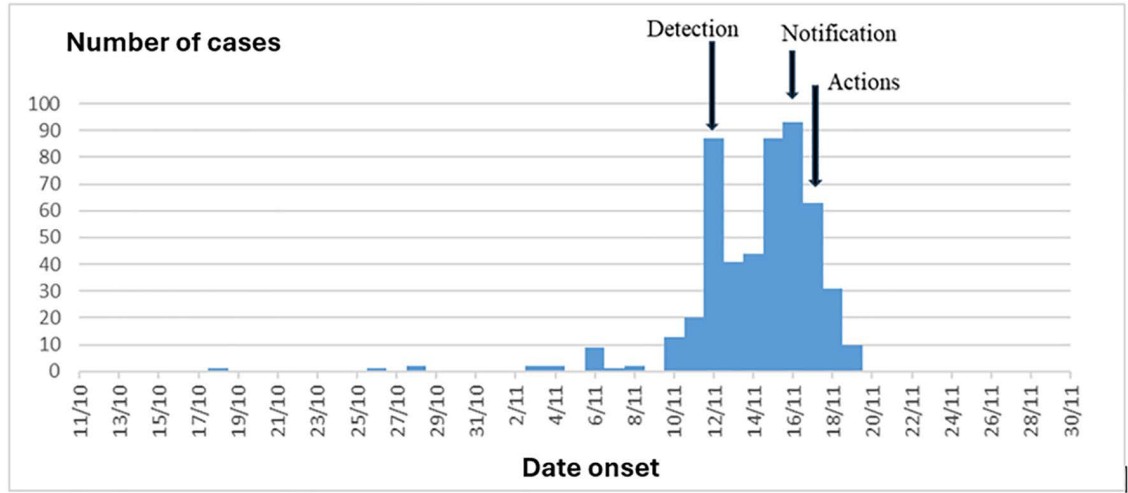

**Fig 2. Temporal evolution of the acute dermatosis outbreak among fishermen in Senegal.**

### Spatial distribution of the outbreak

The outbreak predominantly affects populations originating from the Senegalese coastline in coastal areas known for their fishing activities (Fig 3).

### Socio-demographic characteristics of patients

Mean age of the patients was 22 years old with a standard deviation of 9 years; extremes of 8 and 59 years and un median age of 20 years; the interquartile range (IQR) was [28,16]. The most representative age was 18 years. All patients were men and fishermen by profession.

### Description of clinical signs

The patients presented with fever (16%), skin pain (100%), cutaneous-mucosal lesions (100%) of vesicles, papules, and ulcers features localized on exposed areas of the body, external genital organs, and oral mucosa with conjunctival hyperemia (Table 1). Other signs noted included: bleeding (2%) and neurological disorders (0.9%). According to the severity of the lesions, simple cases that did not require hospitalization represented 92% of those surveyed, and severe cases 8%.
   Some illustrations of the dermatological lesions are presented in Fig 4.

### Description of bacteriological, virological, anatomopathological, and toxicological results

The investigation team conducted sampling (Fig 5) for the following analyses:

• Bacteriological and virological tests performed on a sample of 10 patients returned negative results, particularly for COVID-19 and viral hemorrhagic fevers.

• Anatomopathological examinations (skin biopsy) showed the presence of homogeneous epidermal necrosis foci without any evidence supporting a viral origin.

• Toxicological examinations of environmental samples collected from nets and marine products in search of chemical pollutants, including polychlorinated biphenyls, polycyclic aromatic hydrocarbons, and pharmaceutical residues, returned negative results. Toxicological tests revealed the presence of the marine dinoflagellate *Vulcanodinium rugosum* and its associated cyclic imine toxins.

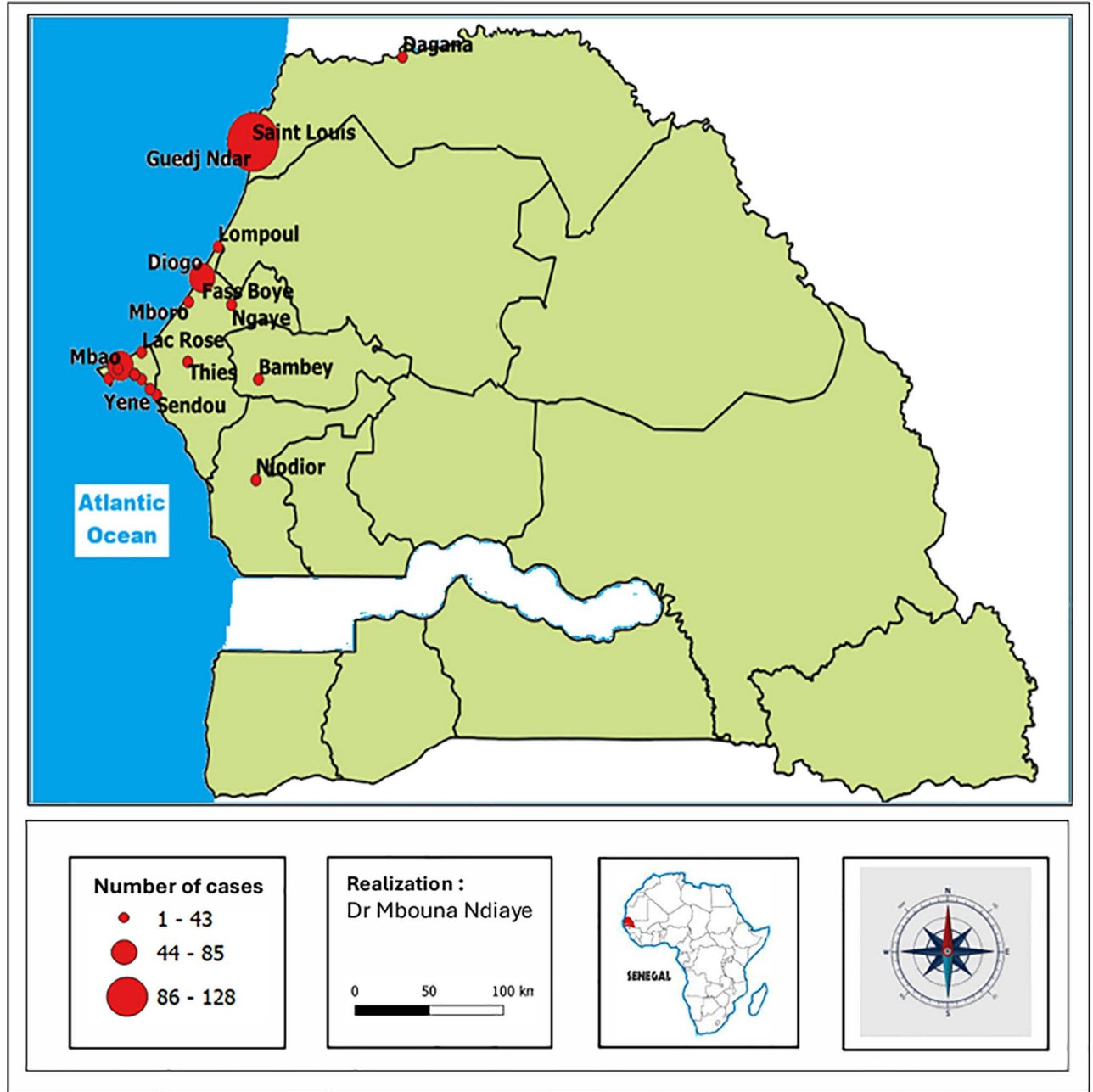

**Fig 3. Cartography of acute dermatoses among fishermen in Senegal according to their origin.**

## Investigation of exposure factors

The notion of a sea voyage within 24–48 hours before the onset of the disease was found in 92% of cases. Additionally, 3% had not undertaken a sea voyage. For some (5%), the notion of a voyage was not specified during the investigation. Regarding the type of equipment used to catch fish, it was mainly gillnets (99.8%) and lines (0.2%); no fisherman had used purse seines. In all cases, it involved artisanal fishing equipment used by a profession operating in an informal setting. The majority of cases (74%) did not have complete personal protective equipment during work (hood, tarp, etc.).

**Table 1. Distribution of clinical signs during the acute dermatosis outbreak among fishermen in Senegal.**

| Signs | Number (n = 555) | Percentage (%) | 95% CI (Lower – Upper) |
|---|---|---|---|
| Skin pain | 555 | 100 | 99.3 – 100 |
| Cutaneous-mucosal lesions | 555 | 100 | 99.3 – 100 |
| Pruritus | 209 | 37.7 | 33.6 – 41.7 |
| Ocular lesions | 170 | 30.6 | 26.8 – 34.5 |
| Fever | 89 | 16.0 | 13.0 – 19.1 |
| Headaches | 58 | 10.5 | 7.9 – 13.0 |
| Myalgia | 36 | 6.5 | 4.4 – 8.5 |
| Bleeding | 11 | 2.0 | 0.8 – 3.1 |
| Neurological disorders | 5 | 0.9 | 0.1 – 1.7 |

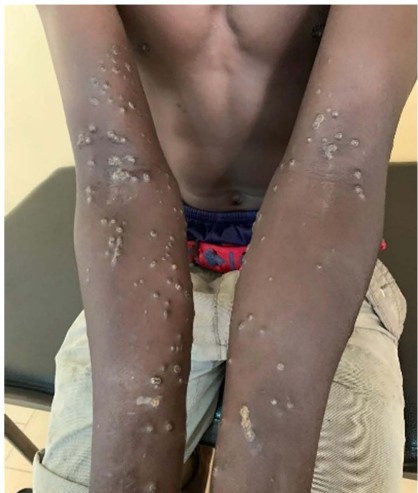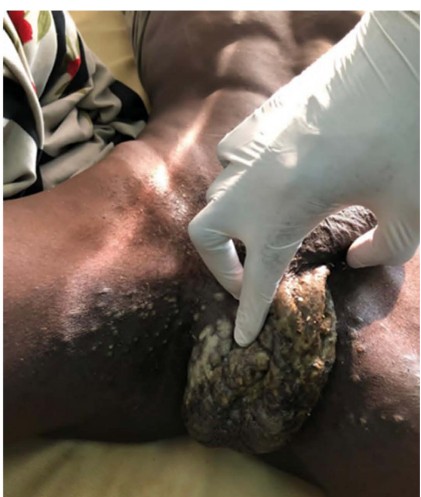

**Fig 4. Illustration of cutaneous-mucosal lesions among fishermen in Senegal.**

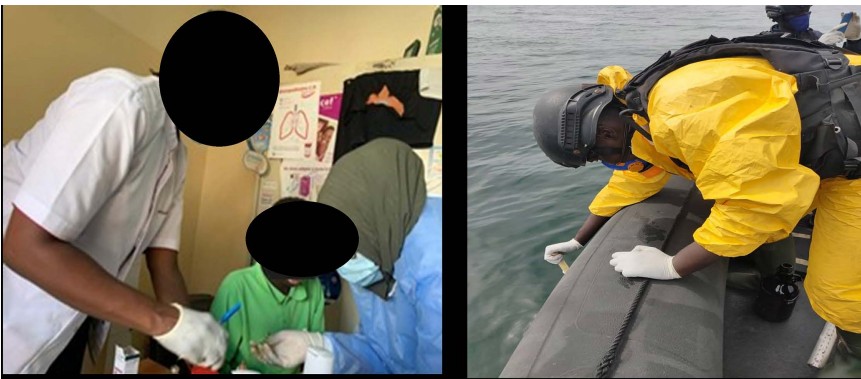

**Fig 5. Investigation (samples) during dermatosis on fishermen, Senegal.**

The proportion of people without complete protection was 83% among those who developed severe forms. People without complete protection were more exposed to severe forms than those with complete personal protective equipment; the difference was not statistically significant (OR = 1.818; 95% CI [0.829 - 3.988]; p = 0.165).

### Consequences of dermatoses

Besides the physical pain caused by the lesions, the disease had significant psychological repercussions on fishermen in the context of the COVID-19 pandemic, with already strained communities and a weakened health system. Stigma and fear caused stress and emotional distress among the patients. The impact on work was significant, with many fishermen unable to work due to the professional nature of the disease, leading to income losses for families dependent on artisanal fishing. Finally, reluctance to consume fish by households, fueled by fear of contamination, negatively impacted the demand for seafood products.

### Strategies and measures taken to control the outbreak

From the outset, an incident management system was established to handle the crisis. It is multidisciplinary and multisectoral, following a "one health" approach with the participation of epidemiologists, dermatologists, environmentalists, toxicologists, bacteriologists, virologists, and anatomopathologists from the human, animal, and environmental health sectors, supported by technical and financial partners. To prepare for any eventuality, an increase in vigilance was ordered at all health service points. Case notification and patient management guidelines (algorithms) were produced. Case definitions and notification forms were shared with health structures. Due to the magnitude of the outbreak and its socio-economic impact, free case management was decreed by the authorities; the psychosocial aspect of the disease was entrusted to the district social services. Indeed, the affected population consisted of artisanal fishermen operating in the informal sector with insufficient health coverage. Given the rapid increase in cases, the spectacular and distressing nature of the symptoms associated with the initially unknown origin of the disease, and the detrimental consequences on the local maritime economy, strong public pressure had to be managed through a specific crisis communication strategy. Simple prevention messages in the local language were disseminated to fishermen to temporarily suspend sea voyages, or alternatively, to wear personal protective equipment (PPE). In case of contact with seawater, it was advised to rinse immediately and thoroughly with fresh water. The use of shea butter constituted an effective protective barrier in addition to PPE. The messages aimed to deconstruct occult and spiritual beliefs about this "mysterious disease" of fishermen, which was widely amplified by the mass media. Regarding curative care, cases were treated based on whether the condition was purely cutaneous (antiseptic) with or without superinfection (+/- antibiotic: amoxicillin clavulanic acid), oral (mouthwash), genital (local care with povidone iodine), or ocular (antiseptic washing solution).

### Lessons learned

The investigation of dermatoses observed among fishermen highlights, one more time, the necessity of a one health approach to unravel the 'mystery' of human diseases of environmental origin, as was the case in this work.

Free healthcare access will always be decreed against an unforeseen large-scale event with a certain impact on the budget of health structures. Therefore, a health system guaranteeing access to care through universal health coverage constitutes the ideal solution, even in the context of epidemics, including the management of primary healthcare.

The incident of acute dermatoses occurring during the COVID-19 pandemic reminds us that one epidemic should not overshadow another. Increasingly, health systems will have to face multiple acute events simultaneously in increasingly complex situations.

## Discussion

This phenomenon is new, meaning that it did not exist in the region before and has just emerged for the first time. To our knowledge, we report the second episode of such outbreak, worldwide after the study conducted in Cuba [8].

## Temporal-spatial evolution

The incident occurred in October-November 2020, covering part of the rainy season. In Senegal [14], in 2020, the rainy season lasted 4.2 months, from June 23 to October 31. In Cuba, the episode of dermatitis was recorded in the summer of 2015, from July to September, during the rainy season [8]. These observations suggest the hypothesis of seasonality of the incident, namely the rainy season.

Temporal analysis shows an exceedance of detection (25 days) and notification (4 days) deadlines, while the action time is deemed acceptable (1 day). Indeed, the use of the 7-1-7 target, according to which countries strive to identify any suspected epidemic within 7 days of its onset, report it to public health authorities by launching investigation and response efforts within one day, and respond effectively within 7 days, can help identify bottlenecks and points of failure, and improve performance [15,16]. The apparent 'decline' in cases observed at the end of the week is explained by the failure in continuity of care at health facilities. In a health structure, continuity of care involves creating and maintaining a complete and coherent care pathway for each person at all times.

The absence of care during periods of minimal service can lead to delays in the detection, notification, and management of acute events.

Spatial analysis revealed a correlation between proximity to the sea and the incidence of dermatoses observed among fishermen. Coastal localities are known for their artisanal fishing activities, which explains the notable concentration of cases among patients from these areas. Likewise, the similar dermatosis described among swimmers in Cuba occurred on two shallow beaches ('El Círculo Juvenil' and 'La Punta'), located very close to each other along the coastline of the city of Cienfuegos, in the central part of the eponymous bay. In our research, the existence of cases outside the coastal area, such as in Bambey, can be explained by the fact that it is an initial address of patients who have moved or relocated to the maritime area.

## Sociodemographic, clinical, and paraclinical characteristics

Our study population was relatively young, with an average age of $22 \pm 9$ years. They were all male and fishermen by profession. This is related to the professional nature of the disease, as artisanal fishing is a physically demanding occupation requiring sustained human effort due to the use of unsophisticated means. Fishing professionals, including those in the agri-food industry, food trade employees (fishmongers, large retailers), and catering professions, are the most affected by contact dermatitis [17].

In the dermatoses of Cienfuegos Bay (Cuba), with a total of 60 patients, 49 cases (81.6%) were men and 11 (18.3%) were women. Children aged 0–14 years represented 79.2% of the cases studied; the average age of the patients was $11.7 \pm 4.6$ years [8]. These differences with our study are explained by the fact that the dermatosis occurred in Cienfuegos among swimmers during a recreational activity, primarily involving children and a population of both sexes.

In our investigation, the absence of fever in the majority of patients and the lack of human-to-human transmission (e.g., male fishermen did not infect women who stayed at home), the negative results of bacteriological and virological tests, and the lack of anatomopathological evidence supporting a viral origin are all arguments pointing towards a non-infectious origin. The technical difficulties and the limited resources for diagnosis were suggestive of conducting toxicological tests abroad, which formally identified the causal agent (*Vulcanodinium rugosum*). The toxin Portimine A produced by *V. rugosum* blocks protein production in cells, resulting in the activation of a powerful immune sensor, the NLRP1 inflammasome, leading to the release of proinflammatory cytokines IL1β/IL18 and cell necrosis. This situation sufficiently demonstrates the importance of a robust and well-equipped infrastructure to effectively conduct comprehensive and rigorous scientific investigations.

## Exposure factors

Maritime travel is not without risk, as revealed by our study, with 92% of the surveyed individuals having been in contact with the sea before the onset of their illness. It is likely that those who did not travel had manipulated contaminated marine

equipment and products at home. Maritime personnel represent the most exposed population in the workforce to the risk of accidents or injuries [2]. Accidents and injuries are ever-present threats, exacerbated by the unpredictable nature of the marine environment. Occupational risk assessment, workplace environment management, and the use of ergonomic and personal protective equipment are essential to reduce these risks [18]. Strict adherence to health safety protocols is fundamental to ensuring safe working conditions and reducing nuisances. Many causes that can explain the expansion of harmful marine dinoflagellate species [19,20]. Maritime activities related to ever-increasing global trade are regularly implicated in the introduction of toxic species into aquatic ecosystems via the ballast waters of ships, which release these waters loaded with vegetative cells or cysts in port areas [21–23]. Proliferation has also been associated with abnormal weather conditions [8]. Marine ecosystems are warming, acidifying, and deoxygenating due to climate change, and these changes are accompanied by increasing impacts of harmful algal blooms [24,25].

## Strategies and measures taken

The transdisciplinary and collaborative approach adopted in our investigation takes into account the complex and multiple links between the health of living beings, biodiversity, and the environment. This integrated approach, which affirms the interdependence of animal and human health and the state of ecosystems, and promotes new methods of disease surveillance and control, is increasingly gaining interest [26–29]. According to the literature, Senegal serves as a model among countries participating in the prestigious Global Health Security Program (GHSP) based on "One Health." Senegal has early on established the program's governing bodies, carried out almost all the activities planned in the roadmap, and had the ingenious idea of housing the program at the highest level of decision-making and coordination of government action, at the Prime Minister's Office, for better multisectoral coordination [30].

During the incident, crisis communication managed the social tension resulting from the socio-economic difficulties of artisanal fishermen. They experienced many days without work due to restrictions related to the COVID-19 pandemic and the dermatosis epidemic. The decrease in income and food insecurity can lead to a deterioration in living conditions, forcing some to seek economic alternatives that are often less safe and less remunerative.

Health systems faced challenges of simultaneous management of the COVID-19 pandemic and the dermatosis outbreak. With human resources already mobilized for the fight against COVID-19, health services had to quickly adapt to respond to the new challenges posed by dermatosis.

The lack of health coverage in the informal sector, such as fishing, constitutes a major constraint to accessing care. Despite the progress made, there are still many obstacles to universal health coverage and its two main objectives: improving access to quality healthcare and reducing the financial burden on families [31].

## Limitations

The study only concerns cases seen in health facilities, while there are surely unseen cases that have remained at home. As the case definition is based on clinical (dermatological examination), a differential diagnosis with other pathologies could be difficult for some atypical cases. Finally, an analytical study (with a comparison group) would have helped better to support the hypotheses, but this was not planned in these conditions of health emergency. Furthermore, the selection of small groups for complementary examinations or qualitative studies is due to the insufficiency of available resources. In addition, the study was unable to precisely measure the dose or duration of exposure to *V. rugosum* toxins during contact at sea.

## Conclusion

The investigation of the acute dermatosis outbreak observed among fishermen in Senegal identified the probable cause as a toxin (portimine A) secreted by a marine microalga (*Vulcanodinium rugosum*).

We implemented control measures to avoid contact with the source through a communication method tailored to the fishermen. Moreover, efforts to provide fishermen with personal protective equipment should be supported, and

awareness should be raised to encourage them to use these effective means of preventing toxic dermatitis and other injuries.

Given the importance of fishing in the economic fabric, in addition to the usual safety aspects (life jacket, compass, flashlight, etc.), it is necessary to consider a new issue: the health safety of workers at sea with epidemic threats.

We also recommend a surveillance mode for specific at-risk populations alongside the existing sentinel site surveillance, as other incidents related to the blue economy are expected in the future, in connection with ecosystem disruptions.

Furthermore, epidemiological surveillance efforts should focus on the effectiveness and efficiency of an early warning system that allows detection within 7 days and rapid notification to the higher level of the health pyramid within 24 hours. It is also crucial to ensure the continuous availability of health services 24/7 to avoid missing cases and causing severe consequences in the management of medical and public health emergencies. Our investigation highlighted the need to improve diagnostic capacities for future similar investigations, given the insufficiency of the technical platform to allow toxicological examinations to be carried out in the country. From a public health perspective, this study highlights how epidemiological surveillance, combined with evidence-informed action and community awareness, can contribute to reducing the impact of outbreaks at the intersection of environmental and occupational health.

## Supporting information

**S1 Appendix. Laboratory Protocols and Validation.**
(DOCX)

**S2 Appendix. Toxin Detection, Organism Identification and Chain of Custody.**
(DOCX)

**S3 Appendix. Psychosocial and Professional Impact.**
(DOCX)

## Acknowledgments

We thank the institutions, organizations and individuals who participated in the study or supported us in this work, in particular the Ministries (human, animal and environmental health), customary, territorial, administrative and health authorities, fishermen and their families.

## Author contributions

**Conceptualization:** Mbouna Ndiaye, Ndeye Magatte Ndiaye, Fatimata Ly, Mamadou Ndiaye, Boly Diop, Alassane Ndiaye, Elhadji Mamadou Ndiaye, Moussa Diallo, Mamadou Fall, Marie Khemesse Ngom Ndiaye.

**Data curation:** Mbouna Ndiaye, Ndeye Magatte Ndiaye, Fatimata Ly, Mamadou Ndiaye, Boly Diop.

**Formal analysis:** Mbouna Ndiaye, Ndeye Magatte Ndiaye, Mamadou Ndiaye, Boly Diop, Diambogne Ndour, Mamadou Fall.

**Investigation:** Mbouna Ndiaye, Ndeye Magatte Ndiaye, Fatimata Ly, Mamadou Ndiaye, Boly Diop, Diambogne Ndour, Alassane Ndiaye, Elhadji Mamadou Ndiaye, Moussa Diallo, Mamadou Fall, Marie Khemesse Ngom Ndiaye.

**Methodology:** Mbouna Ndiaye, Ndeye Magatte Ndiaye, Fatimata Ly, Mamadou Ndiaye, Boly Diop, Diambogne Ndour, Alassane Ndiaye, Elhadji Mamadou Ndiaye, Moussa Diallo, Mamadou Fall, Marie Khemesse Ngom Ndiaye.

**Project administration:** Ndeye Magatte Ndiaye, Boly Diop, Diambogne Ndour, Elhadji Mamadou Ndiaye, Marie Khemesse Ngom Ndiaye.

**Resources:** Ndeye Magatte Ndiaye, Boly Diop, Diambogne Ndour, Elhadji Mamadou Ndiaye, Moussa Diallo, Marie Khemesse Ngom Ndiaye.

**Software:** Mbouna Ndiaye, Mamadou Ndiaye, Alassane Ndiaye.

**Supervision:** Ndeye Magatte Ndiaye, Fatimata Ly, Mamadou Ndiaye, Boly Diop, Diambogne Ndour, Elhadji Mamadou Ndiaye, Moussa Diallo, Mamadou Fall, Marie Khemesse Ngom Ndiaye.

**Validation:** Mbouna Ndiaye, Ndeye Magatte Ndiaye, Fatimata Ly, Mamadou Ndiaye, Boly Diop, Diambogne Ndour, Alassane Ndiaye, Elhadji Mamadou Ndiaye, Mamadou Fall, Marie Khemesse Ngom Ndiaye.

**Visualization:** Mbouna Ndiaye, Fatimata Ly, Boly Diop, Diambogne Ndour, Alassane Ndiaye, Mamadou Fall.

**Writing – original draft:** Mbouna Ndiaye, Ndeye Magatte Ndiaye, Fatimata Ly, Mamadou Ndiaye, Diambogne Ndour, Alassane Ndiaye, Elhadji Mamadou Ndiaye, Moussa Diallo, Mamadou Fall, Marie Khemesse Ngom Ndiaye.

**Writing – review & editing:** Mbouna Ndiaye, Ndeye Magatte Ndiaye, Fatimata Ly, Mamadou Ndiaye, Boly Diop, Diambogne Ndour, Alassane Ndiaye, Elhadji Mamadou Ndiaye, Moussa Diallo, Mamadou Fall, Marie Khemesse Ngom Ndiaye.

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
