## [Decision Letter · Decision Letter 0]

26 Aug 2025

PGPH-D-25-01888

Investigation of an outbreak of unusual acute dermatoses of algal origin among fishermen in Senegal

Dear Dr. Mbouna Ndiaye,

Thank you for submitting your manuscript to PLOS Global Public Health. After careful consideration, we feel that it has merit but does not fully meet PLOS Global Public Health’s publication criteria as it currently stands. Therefore, we invite you to submit a revised version of the manuscript that addresses the points raised during the review process.

We look forward to receiving your revised manuscript.

Kind regards,

Muhammad Asaduzzaman, MD MPH MPhil

Academic Editor

Journal Requirements:

1. We have amended your Competing Interest statement to comply with journal style. We kindly ask that you double check the statement and let us know if anything is incorrect.

3. Some material included in your submission may be copyrighted. According to PLOS’s copyright policy, authors who use figures or other material (e.g., graphics, clipart, maps) from another author or copyright holder must demonstrate or obtain permission to publish this material under the Creative Commons Attribution 4.0 International (CC BY 4.0) License used by PLOS journals. Please closely review the details of PLOS’s copyright requirements here: PLOS Licenses and Copyright. If you need to request permissions from a copyright holder, you may use PLOS's Copyright Content Permission form.

Potential Copyright Issues:

a) Figure 1: please (a) provide a direct link to the base layer of the map (i.e., the country or region border shape) and ensure this is also included in the figure legend; and (b) provide a link to the terms of use / license information for the base layer image or shapefile. We cannot publish proprietary or copyrighted maps (e.g. Google Maps, Mapquest) and the terms of use for your map base layer must be compatible with our CC-BY 4.0 license.

b) Figures 3 and 4 includes an image of an identifiable person. Please provide written confirmation or release forms, signed by the subject(s) (or their parent/legally authorized guardian), giving permission to be photographed and to have their images published under our CC-BY 4.0 license.

Otherwise, we kindly request that you remove the photograph.

Additional Editor Comments (if provided):

Reviewers' comments:

Reviewer's Responses to Questions

**Comments to the Author**

1. Does this manuscript meet PLOS Global Public Health’s publication criteria? Is the manuscript technically sound, and do the data support the conclusions? The manuscript must describe methodologically and ethically rigorous research with conclusions that are appropriately drawn based on the data presented.? Is the manuscript technically sound, and do the data support the conclusions? The manuscript must describe methodologically and ethically rigorous research with conclusions that are appropriately drawn based on the data presented.

Reviewer #1: Yes

Reviewer #2: Yes

2. Has the statistical analysis been performed appropriately and rigorously?

Reviewer #1: Yes

Reviewer #2: Yes

3. Have the authors made all data underlying the findings in their manuscript fully available (please refer to the Data Availability Statement at the start of the manuscript PDF file)?

The PLOS Data policy requires authors to make all data underlying the findings described in their manuscript fully available without restriction, with rare exception. The data should be provided as part of the manuscript or its supporting information, or deposited to a public repository. For example, in addition to summary statistics, the data points behind means, medians and variance measures should be available. If there are restrictions on publicly sharing data—e.g. participant privacy or use of data from a third party—those must be specified.requires authors to make all data underlying the findings described in their manuscript fully available without restriction, with rare exception. The data should be provided as part of the manuscript or its supporting information, or deposited to a public repository. For example, in addition to summary statistics, the data points behind means, medians and variance measures should be available. If there are restrictions on publicly sharing data—e.g. participant privacy or use of data from a third party—those must be specified.

Reviewer #1: No

Reviewer #2: No

4. Is the manuscript presented in an intelligible fashion and written in standard English?

Reviewer #1: Yes

Reviewer #2: Yes

Reviewer #1: 3: Authors have stated Data Availability as None. They should explain why the data from this study cannot be made available publicly, for example, due to patient confidentiality or government data sharing restrictions.

Reviewer #2: I would like to thank Plos for giving me the honor of being among the reviewers of this article. I also congratulate the authors for taking an interest in an unusual phenomenon that is very poorly documented, both globally and in the African context. The evidence presented in this article provides valuable insights that could inform preventive strategies for the disease among fishermen. Nevertheless, below are a few observations, most of which are minor or simply matters of curiosity.

L56 to 71, Minor point of view: this section focuses excessively on general topics instead of quickly addressing the specific subject of interest, which is marine algae dermatoses discussed in the article. While beginning with general information can be useful before moving on to specifics, it's important to remain concise by only briefly mentioning these general aspects in the introduction. Unfortunately, this section does not achieve that balance.

The paper shows that the extent, symptoms, and causes of the phenomenon were unknown when the study was designed (L90-93), making it an exploratory investigation. Given the alerts that prompted this research, paragraphs L84-86 should mention several possible causes instead of just one confirmed cause. This implies that the introduction was written after the analysis, which undermines suspense until the results are revealed (L234). The general aspects I noted earlier—like pollution, chemical agents, allergens, and bacterial agents—should be presented as hypotheses. If marine algal dermatoses were the only considered cause, it should be clearly stated in the introduction, but multiple hypotheses should also be included to guide the reader on the authors' investigative direction.

L77-83: Shouldn't this section be combined into a single paragraph with lines 74-76?

L31, L129: Question: How is this study multicenter? It would be helpful to provide additional details to explain the use of this term.

L142-152: It would be beneficial to clarify whether all participants were eligible for various parts of the research or if there was a specific eligibility algorithm for each section.

Important details are missing in the methodology section. There is a gap regarding sampling. Key aspects to clarify include the estimated sample size and participants selection process. How many subjects were attended to be included? Were all healthcare facilities in the study region selected, or only some? If only some, what criteria were used? Were healthcare facilities the only way of identifying participants, or were there other methods? Once identified in healthcare facilities, how were these participants approached, especially for patients not admitted in hospitalisation ? Were all eligible cases included, or were some identified but not part of the study for specific reasons?

L188-196: It is important to clarify whether the phenomenon is new, meaning it did not previously exist in the region and has just emerged, or if it existed before but on a smaller scale. Neither this paragraph nor those between lines 87-93 or 299-300 provide specific information.

L210 “All patients in the study were men who worked as fishermen”. The lack of detail on the sampling technique raises concerns about potential selection or information bias. The definition of the study population (lines 131-137) also raises questions, particularly given that this profession is predominantly male, as indicated by the criteria stating, “...who had been at sea in the five days prior to the onset of symptoms, ...” Additionally, the skin lesions in the case definition may exclude other skin diseases affecting both genders. In an epidemiological study, the inclusion criteria should be sensitive, while the diagnostic criteria must be specific. It is important to distinguish between the characteristics of the evaluated subjects (ideally including both men and women) and those of confirmed cases (which may consist only of men). I recommend that the authors provide a flow chart detailing the evaluated individuals, including the 555 declared ill, while specifying the gender breakdown and reasons for exclusion at each stage.

L224. These photos likely show lesions that were still active during the investigation. Given the 0% mortality rate, what were the possible outcomes? If recovery was one, what was the average illness duration? Were there complications, and how were they treated? If data is available, it would be useful to compile these results into a single section (e.g., L283-286) for a clearer overview of clinical aspects (L211). Additionally, the authors should clarify in the methodology how the differential diagnosis of dermatoses was made and who was responsible for it.

L228 I have part of the answer to my question: “Were all participants eligible for all parts of the study?” It appears that they were not. Therefore, my observation remains valid: this aspect should be clearly specified in the methodology. Additionally, the criteria for selecting these 10 patients or for determining eligibility for any part of the study should also be outlined.

Regarding lines 255-262 and lines 263-286: Are there any figures or tables to support these results? Is this section based on a qualitative approach? If so, the methodology should specify that both quantitative and qualitative approaches were used.

**Do you want your identity to be public for this peer review?** For information about this choice, including consent withdrawal, please see our Privacy Policy..

Reviewer #1: **Yes:** Novil WijesekaraNovil WijesekaraNovil WijesekaraNovil Wijesekara

Reviewer #2: No

---

## [Decision Letter · Decision Letter 1]

14 Oct 2025

PGPH-D-25-01888R1

Investigation of an outbreak of unusual acute dermatoses of algal origin among fishermen in Senegal

Dear Dr. Mbouna Ndiaye

Thank you for submitting your manuscript to PLOS Global Public Health. After careful consideration, we feel that it has merit but does not fully meet PLOS Global Public Health’s publication criteria as it currently stands. Therefore, we invite you to submit a revised version of the manuscript that addresses the points raised during the review process.

We look forward to receiving your revised manuscript.

Kind regards,

Titilayo Abike Olaoye, PhD

Academic Editor

Journal Requirements:

Additional Editor Comments (if provided):

General Comments: The study addresses a highly relevant and novel outbreak of acute dermatoses linked to algal toxins among fishermen in Senegal. The integration of field epidemiology, laboratory analyses, and One Health perspectives makes it potentially valuable for publication. However, the manuscript still requires substantial revisions to improve clarity, methodological transparency, and interpretive depth. Specific issues are outlined below.

Specific Comments

• Consider revising the title to: “Investigation of an outbreak of acute algal-associated dermatoses among artisanal fishermen in Senegal: A One Health field investigation.”

• Add a sentence summarizing public health implications in the Conclusion.

• Case Ascertainment and Sampling: Please include a flow diagram showing how the 555 cases were identified, screened, and included/excluded in different components (epidemiologic, laboratory, qualitative). Clarify whether cases were identified only through health facilities or also through community-based outreach. Discuss how the case definition might have excluded atypical or mild cases and introduce this as a limitation.

• Population Denominator and Attack Rate: Justify the use of the regional population as the denominator for the attack rate. Provide an alternative calculation using the exposed fishing population (e.g., 10,200 fishermen) and report both rates. Delete the numerator/denominator values explicitly in the Results section, and report only the percentages.

• Selection Criteria for Laboratory Testing: Clearly describe how the 10 bacterio-virology samples, 4 biopsies, and 8 environmental samples were selected. Indicate whether these were randomly chosen, convenience-based, or severity-driven. Provide details of the laboratory protocols (sample preparation, instruments, analytical methods, detection limits, quality controls, inter-laboratory validation with IFREMER).

• Toxicology and Identification of Vulcanodinium rugosum: Expand on the methods for toxin detection and organism identification: specify techniques (e.g., LC-MS/MS, microscopy, PCR), reference standards, and validation procedures. Clarify the chain of custody and storage conditions for samples.

• Interpret results cautiously—rephrase causal claims to “probable algal origin” or “evidence consistent with” rather than definitive causation.

• Statistical and Epidemiological Reporting: Ensure all tables and text include 95% confidence intervals and p-values for effect estimates (e.g., PPE use vs. severity). Verify consistency between numbers in Abstract, Results, and Tables. If available, report disease duration, hospitalization rate, and clinical outcomes; otherwise, state that data were not recorded.

• Qualitative Component: Include a brief Qualitative Methods subsection: describe participant selection, interview/focus group process, data transcription, coding, and thematic analysis method.

• Present a short table summarizing key themes or illustrative quotes related to psychosocial and occupational impact.

• Discussion and Interpretation: Avoid repetition of results. Deepen analysis by explaining why the outbreak occurred, drawing parallels with similar events (e.g., Cuba). Discuss possible biological mechanisms of portimine A and environmental triggers.

• Acknowledge and elaborate on study limitations (sampling bias, lack of exposure quantification, recall bias, absence of control group).

• Ethical Considerations and Image Permissions: Provide full details of ethical approval (IRB name, number, and date). State how informed consent was obtained (and parental consent if minors are included). Confirm that patient images were obscured/anonymized and used with consent or appropriately removed.

• Data Availability:-Provide a complete Data Availability Statement as required by PLOS: State where the dataset and supporting files can be accessed (repository name, DOI/URL). If access is restricted, specify the reason and contact person for data requests.

• Tables, Figures, and Formatting: Ensure that every table has clear titles; defined abbreviations; Correct totals and percentages; Appropriate footnotes explaining subsamples (e.g., biopsy, bacteriology); Include figure captions stating data sources, map copyrights, and consent for images. Standardize formatting throughout (fonts, heading levels, spacing).

• Language and Style: Simplify long sentences, especially in the Introduction and Discussion. Ensure consistent tense use (Methods in past tense; Results in present; Discussion in past/present).

• References: Update literature to include recent (2020–2024) works on harmful algal blooms, marine toxin exposure, and dermal toxicity. Standardize all references according to PLOS style (authors, year, title, journal, DOI). Verify that all in-text citations correspond to the reference list.

Reviewers' comments:

Reviewer's Responses to Questions

**Comments to the Author**

Reviewer #1: All comments have been addressed

publication criteria? Is the manuscript technically sound, and do the data support the conclusions? The manuscript must describe methodologically and ethically rigorous research with conclusions that are appropriately drawn based on the data presented.? Is the manuscript technically sound, and do the data support the conclusions? The manuscript must describe methodologically and ethically rigorous research with conclusions that are appropriately drawn based on the data presented.

Reviewer #1: Yes

3. Has the statistical analysis been performed appropriately and rigorously?

Reviewer #1: Yes

4. Have the authors made all data underlying the findings in their manuscript fully available (please refer to the Data Availability Statement at the start of the manuscript PDF file)?

The PLOS Data policy requires authors to make all data underlying the findings described in their manuscript fully available without restriction, with rare exception. The data should be provided as part of the manuscript or its supporting information, or deposited to a public repository. For example, in addition to summary statistics, the data points behind means, medians and variance measures should be available. If there are restrictions on publicly sharing data—e.g. participant privacy or use of data from a third party—those must be specified.requires authors to make all data underlying the findings described in their manuscript fully available without restriction, with rare exception. The data should be provided as part of the manuscript or its supporting information, or deposited to a public repository. For example, in addition to summary statistics, the data points behind means, medians and variance measures should be available. If there are restrictions on publicly sharing data—e.g. participant privacy or use of data from a third party—those must be specified.

Reviewer #1: Yes

5. Is the manuscript presented in an intelligible fashion and written in standard English?

Reviewer #1: Yes

Reviewer #1: Great work in revising the manuscript!

**Do you want your identity to be public for this peer review?** For information about this choice, including consent withdrawal, please see our Privacy Policy..

Reviewer #1: No

---

## [Decision Letter · Decision Letter 2]

28 Jan 2026

PGPH-D-25-01888R2

Investigation of an outbreak of acute algal-associated dermatoses among artisanal fishermen in Senegal: a one health approach

Dear Dr. Mbouna Ndiaye

Thank you for submitting your manuscript to PLOS Global Public Health. After careful consideration, we feel that it has merit but does not fully meet PLOS Global Public Health’s publication criteria as it currently stands. Therefore, we invite you to submit a revised version of the manuscript that addresses the points raised during the review process.

We look forward to receiving your revised manuscript.

Kind regards,

Titilayo Abike Olaoye, PhD

Academic Editor

Journal Requirements:

Additional Editor Comments (if provided):

Reviewers' comments:

Reviewer's Responses to Questions

**Comments to the Author**

Reviewer #1: All comments have been addressed

Reviewer #2: (No Response)

publication criteria? Is the manuscript technically sound, and do the data support the conclusions? The manuscript must describe methodologically and ethically rigorous research with conclusions that are appropriately drawn based on the data presented.? Is the manuscript technically sound, and do the data support the conclusions? The manuscript must describe methodologically and ethically rigorous research with conclusions that are appropriately drawn based on the data presented.

Reviewer #1: Yes

Reviewer #2: Yes

3. Has the statistical analysis been performed appropriately and rigorously?

Reviewer #1: Yes

Reviewer #2: I don't know

4. Have the authors made all data underlying the findings in their manuscript fully available (please refer to the Data Availability Statement at the start of the manuscript PDF file)?

The PLOS Data policy requires authors to make all data underlying the findings described in their manuscript fully available without restriction, with rare exception. The data should be provided as part of the manuscript or its supporting information, or deposited to a public repository. For example, in addition to summary statistics, the data points behind means, medians and variance measures should be available. If there are restrictions on publicly sharing data—e.g. participant privacy or use of data from a third party—those must be specified.requires authors to make all data underlying the findings described in their manuscript fully available without restriction, with rare exception. The data should be provided as part of the manuscript or its supporting information, or deposited to a public repository. For example, in addition to summary statistics, the data points behind means, medians and variance measures should be available. If there are restrictions on publicly sharing data—e.g. participant privacy or use of data from a third party—those must be specified.

Reviewer #1: No

Reviewer #2: Yes

5. Is the manuscript presented in an intelligible fashion and written in standard English?

Reviewer #1: Yes

Reviewer #2: Yes

Reviewer #1: Dear authors,

Congratulations on improving the manuscript. I see some of the reviewer comments still have not been fully addressed. Please address each of them carefully, as it would help improve your manuscript. If you are not willing to do any of the recommended revisions, please justify the reason. Good luck with the revisions!

Reviewer #2: The improvements are evident, and I would like to congratulate the authors on their efforts. However, However, some improvements have raised new questions and concerns that we wish to address with the authors.

1°) Lines 27-30 : The background in the abstract suggests that the causative agent (Vulcanodinium rugosum) was already known at the start of the study, which raises issues of consistency and chronology for me. In my opinion, the name of the causative agent should not appear in the background unless it is in the form of a hypothesis. Rather, it should appear as one of the results of the investigations. I also think that identifying the causative agent should be added to the study's aim. I suggest that the authors delete the first sentence of the abstract or replace it with the sentence found on lines 65 to 67 of the introduction section. It would be useful to specify the nature of the alert for the mysterious disease as a dermatosis probably caused by fungi.

I suggest this, and the authors must be free to improve: In November 2020, an alert for a “mysterious skin disease” among fishermen was issued. Fishermen are particularly vulnerable to dermatoses, especially those caused by algae, due to their constant contact with seawater, fish, crustaceans, and fishing equipment that may contain harmful agents. The study aimed to investigate the alert, to identify the causative agent, and to propose preventive and control measures.

2°) Lines 31-34 In my opinion, the methodology is still rather poor. It should provide a little more detail on sampling (study population, inclusion criteria, selection), data collection (where, how, etc.), sampling for bacteriological analyses, etc. The section should state that both qualitative and quantitative approaches were used.

Line 47: why “probable” when algae has been isolated?

In the methodology section, participants for the qualitative approach (20), bacteriological test (10), and biopsy (4) were randomly selected. The authors should explain how they decided on these sample sizes and discuss any limitations that come with using small groups.

Lines 181-183: The authors state that the chi-square test was used to estimate the odds ratio (OR). However, only one OR is reported in the results section (lines 257-258). It is recommended that all results from the bivariate analyses be presented in a table. Out of curiosity: why did they not consider going further with multivariate analyses?

In the methodology section, participants for the qualitative approach (20), bacteriological test (10), and biopsy (4) were randomly selected. The authors should explain how they determined these sample sizes and discuss any limitations associated with using small groups.

**Do you want your identity to be public for this peer review?** For information about this choice, including consent withdrawal, please see our Privacy Policy..

Reviewer #1: No

Reviewer #2: No

---

## [Editor Report · Decision Letter 3]

24 Feb 2026

PGPH-D-25-01888R3

Investigation of an outbreak of acute algal-associated dermatoses among artisanal fishermen in Senegal: a one health approach

Dear Dr. Mbouna Ndiaye

Thank you for submitting your manuscript to PLOS Global Public Health. After careful consideration, we feel that it has merit but does not fully meet PLOS Global Public Health’s publication criteria as it currently stands. Therefore, we invite you to submit a revised version of the manuscript that addresses the points raised during the review process.

Please submit your revised manuscript by 17th March, 2026 If you will need more time than this to complete your revisions, please reply to this message or contact the journal office at globalpubhealth@plos.org. Please include the following items when submitting your revised manuscript:

We look forward to receiving your revised manuscript.

Kind regards,

Titilayo Abike Olaoye, PhD

Academic Editor
---

## [Decision Letter · Decision Letter 4]

1 Apr 2026

Investigation of an outbreak of acute algal-associated dermatoses among artisanal fishermen in Senegal: a one health approach

PGPH-D-25-01888R4

Dear Dr Mbouna Ndiaye

We are pleased to inform you that your manuscript 'Investigation of an outbreak of acute algal-associated dermatoses among artisanal fishermen in Senegal: a one health approach' has been provisionally accepted for publication in PLOS Global Public Health.

Best regards,

Titilayo Abike Olaoye, PhD

Academic Editor

Reviewer Comments (if any, and for reference):

Reviewer's Responses to Questions

**Comments to the Author**

Reviewer #1: All comments have been addressed

publication criteria? Is the manuscript technically sound, and do the data support the conclusions? The manuscript must describe methodologically and ethically rigorous research with conclusions that are appropriately drawn based on the data presented.? Is the manuscript technically sound, and do the data support the conclusions? The manuscript must describe methodologically and ethically rigorous research with conclusions that are appropriately drawn based on the data presented.

Reviewer #1: Yes

3. Has the statistical analysis been performed appropriately and rigorously?

Reviewer #1: Yes

4. Have the authors made all data underlying the findings in their manuscript fully available (please refer to the Data Availability Statement at the start of the manuscript PDF file)?

The PLOS Data policy requires authors to make all data underlying the findings described in their manuscript fully available without restriction, with rare exception. The data should be provided as part of the manuscript or its supporting information, or deposited to a public repository. For example, in addition to summary statistics, the data points behind means, medians and variance measures should be available. If there are restrictions on publicly sharing data—e.g. participant privacy or use of data from a third party—those must be specified.requires authors to make all data underlying the findings described in their manuscript fully available without restriction, with rare exception. The data should be provided as part of the manuscript or its supporting information, or deposited to a public repository. For example, in addition to summary statistics, the data points behind means, medians and variance measures should be available. If there are restrictions on publicly sharing data—e.g. participant privacy or use of data from a third party—those must be specified.

Reviewer #1: Yes

5. Is the manuscript presented in an intelligible fashion and written in standard English?

Reviewer #1: Yes

Reviewer #1: Thanks a lot for addressing the comments provided.

**Do you want your identity to be public for this peer review?** For information about this choice, including consent withdrawal, please see our Privacy Policy..

Reviewer #1: No
